



# Amino acid racemization in Quaternary foraminifera from the Yermak Plateau

Gabriel West[1], Darrell S. Kaufman[2], Francesco Muschitiello[3], Matthias Forwick[4], Jens
Matthiessen[5], Jutta Wollenburg[5], and Matt O'Regan[1]

[1]Department of Geological Sciences, Stockholm University, SE-10691 Stockholm, Sweden

[2]School of Earth and Sustainability, Northern Arizona University, Flagstaff, AZ 86011, USA

[3]Department of Geography, University of Cambridge, Cambridge, CB2 3EN, UK

[4]Department of Geosciences, UiT The Arctic University of Norway, N-9037 Tromsø, Norway

[5]Alfred-Wegener Institute for Polar and Marine Research, D-27570 Bremerhaven, Germany

*Correspondence to*: Gabriel West (gabriel.west@geo.su.se)

**Abstract**

Amino acid racemization (AAR) geochronology is a powerful tool for dating Quaternary marine sediments across the globe, yet its application to Arctic Ocean sediments has been limited. Anomalous rates of AAR in foraminifera from the central Arctic were reported in previously published studies, indicating that either the rate of racemization is higher in this area, or inaccurate age models were used to constrain the sediment ages. This study investigates racemization rates in foraminifera from three well-dated sediment cores taken from the Yermak Plateau during the 2015 TRANSSIZ Expedition on RV Polarstern. D and L isomers of the amino acids, aspartic acid (Asp) and glutamic acid (Glu), were separated in samples of the planktic foraminifera, *Neogloboquadrina pachyderma* and the benthic species, *Cassidulina neoteretis* to quantify the extent of racemization. In total, 241 subsamples were analysed, extending back to Marine Isotope Stage (MIS) 7. Two previously published power functions, which relate the extent of racemization of Asp and Glu in foraminifera to sample age are revisited, and a comparison is made between the ages predicted by these calibrated age equations and independent geochronological constraints available for the cores. Our analyses reveal an excellent match between ages predicted by a global compilation of racemization rates for *N. pachyderma*, and confirm that a proposed Arctic-specific calibration curve is not applicable at the Yermak Plateau. These results generally support the rates of AAR determined for other cold bottom water sites, and further highlight the anomalous nature of the purportedly high rate of racemization indicated by previous analyses of central Arctic sediments.

Keywords:

    amino acid racemization

    Quaternary geochronology

    foraminifera

    marine sediments

    Arctic Ocean



## 1   Introduction

Dating Quaternary marine sediments from the Arctic Ocean has been a long-standing problem, and a plethora of studies
(e.g. Backman et al., 2004; Stein, 2011; Alexanderson et al., 2014) highlight the challenges of establishing firm
chronologies for these sediments. Assigning ages for the various lithostratigprahic units in Arctic Ocean sediments is,
however, of paramount importance as the development of accurate age models is key to contextualize Arctic
palaeoceanography within Earth's climate system.

Amino acid racemization (AAR) geochronology was first applied specifically to Arctic Ocean sediments in the
pioneering studies of Sejrup et al. (1984), and later that of Macko and Aksu (1986). However, due to the limitations –
associated with small sample size and low temporal resolution – of these early studies, the application of AAR
chronology for Arctic Ocean sediments decreased in the following years. However, major developments in AAR
dating, such as the increased use of the more rapidly racemizing aspartic acid over isoleucine (e.g. Goodfriend et al.,
1996), application of the reverse phase liquid chromatography to determine amino D/L values in small samples
(Kaufman and Manley, 1998) coupled with improvements in Arctic Ocean sediment chronostratigraphies (e.g.
Jakobsson et al., 2001; Backman et al., 2008; O'Regan et al., 2008) reignited the interest in AAR studies during the
past decade.

Kaufman et al. (2008) related the extent of AAR in the polar foraminifera species *Neogloboquadrina pachyderma* (*N.
pachyderma*), collected from various parts of the Arctic Ocean (Mendeleev Ridge, Lomonosov Ridge, Northwind
Ridge), to sediment ages by developing a calibrated age equation for the past 150 ka. This was followed by the study
of Kaufman et al. (2013) who reported the results of AAR analysis of multiple foraminifera species from sediment
cores taken in the Arctic, Atlantic and Pacific Oceans, and defined a general rate of racemization at deep-sea sites.
Both models relied on calibration of the rate of racemization of aspartic (Asp) and glutamic (Glu) acids by independent
dating methods, such as radiocarbon dating and correlations  with the orbitally tuned, stacked oxygen isotope record
(Martinson et al., 1987), yet for the past 150 ka, these two age equations produced significantly different ages for a
given D/L value. The global compilation of AAR showed that the rate of racemization in the Arctic Ocean was higher
than expected in samples of *N. pachyderma* older than 35 ka, and this could not be explained by differences in the rate
of AAR in *N. pachyderma* relative to other species. It remains to be seen whether the observed rates of racemization
are anomalously high compared to other ocean basins, or alternatively, the age constraints for the studied Arctic Ocean
cores are inaccurate. Sedimentary sequences with robust age control and satisfactory preservation of carbonate
microfossils from the Arctic are required to further investigate this question.

Here we present the results of amino acid racemization analysis of planktic (*N. pachyderma*) and benthic (*C. neoteretis*)
foraminifera from samples of three well-dated sediment cores from the Yermak Plateau. The rates of racemization in
the two species are compared, and the Arctic specific (Kaufman et al., 2008) and global (Kaufman et al., 2013)
calibrated age equations are used to estimate the ages of the samples. The calculated ages are compared with those
derived from age models constructed by combining [14]C dating, oxygen isotope stratigraphy, and the correlation of
environmental magnetic properties to the global benthic oxygen isotope record.



## 2   Materials and methods

### 2.1   Sample material and study area


The analysed foraminifera samples were taken from three sediment cores (PS92/0039-2, PS92/0045-2 and PS92/54-1) recovered from the Yermak Plateau during the TRANSSIZ (TRansitions in the Arctic Seasonal Sea Ice Zone) expedition (May 19 to June 28, 2015) on the research icebreaker *RV Polarstern* (PS92) (Table 1 and Figure 1).


The Yermak Plateau, located north of the Svalbard archipelago, lies at the gateway to the central Arctic Ocean, in an area characterized by the interaction between Atlantic and Arctic waters. The cold surface layer is dominated by the polar foraminifera *N. pachyderma*, which has highest abundances in the top 200 m (Carstens and Wefer, 1992; Carstens et al., 1997; Pados and Spielhagen, 2014; Greco et al., 2019). *Cassidulina neoteretis* is one of the dominant benthic species in sediments from the Yermak Plateau (Bergsten, 1994; Wollenburg and Mackensen,

1998; Wollenburg et al., 2004) and its abundance is often associated with the presence of Atlantic water (Polyak and Solheim, 1994; Slubowska et al., 2005).

### 2.2   Sample ages and quantification of age model uncertainties


The age model of core PS92/39-2 was initially developed by Kremer et al. (2018) who used a combination of AMS [14]C dates, sedimentological correlations, and the occurrence of the benthic foraminifera *Pullenia bulloides* as a stratigraphic marker for event 5.1 (~81 ka) in the polar North Atlantic (Haake and Pflaumann, 1989) and Arctic Ocean (Wollenburg et al., 2001). This age model was further developed by Wiers et al. (2019) by recalibrating the [14]C dates and applying a correlation between environmental magnetic parameters (ratio of anhysteretic remanent susceptibility and bulk magnetic susceptibility) with the global stacked benthic foraminifera $\delta^{18}O_{LR}$ record of Lisiecki and Raymo

(2005). Wiers et al. (2019) extended this method to cores PS92/45-2 and PS92/54-1 to provide age models for the cores (Figure 2). Ability to correlate environmental magnetic parameters to the global benthic $\delta^{18}O$ record in late Quaternary sediments from the Yermak Plateau was first proposed by Xuan et al. (2012), who also utilised [14]C dating and oxygen isotope measurements on planktic foraminifera.


To more rigorously provide a measure of the uncertainty inherent to the correlation-based age models, we applied an automated Monte Carlo algorithm for proxy-to-proxy stratigraphical alignment based on the assumption that changes in magnetic parameters (kARM/k) in our marine cores and variations in the $\delta^{18}O_{LR}$ stack are virtually synchronous. The algorithm used here consists of a modified and improved routine that builds upon previous work by Malinverno

et al. (2013) and Muschitiello et al. (2015; 2016), and that has been widely used for synchronization of different paleoclimate archives (e.g. Wohlfarth et al., 2018; Muschitiello et al., 2019). The method was designed to obtain a sample of optimal alignment functions that relates depth in one record to age (or depth) in another. The median of the samples gives the best correlation and their variability provides a measure of the uncertainty associated with the alignment, whereby the alignment function will have larger uncertainties where the match between the proxy timeseries

is poorly constrained. Further details on adjustment of the age-depth model of Wiers et al. (2019) are contained in Appendix A.



### 2.3 Sample preparation and analytical procedure

Individual specimens of the foraminifera species *N. pachyderma* and *C. neoteretis* were picked from the > 63 μm fraction of wet-sieved, oven-dried (4 hours at 30 °C), 2-cm-thick slices of sediment samples from core depths in which tests of the two species were abundant. This resulted in a total of 10, 10 and 5 sampled stratigraphic levels in cores PS92/39-2, PS92/45-2 and PS92/54-1, respectively. Seven core depths were sampled for both *N. pachyderma* and *C. neoteretis* in cores PS92/45-2 and PS92/54-1. Core PS92/39-2 was only sampled for *N. pachyderma*.


The foraminifera tests were first cleaned by sonication (approximately 1 s) in a bath sonicator, then were immersed in 1 ml of 3% $H_2O_2$ for 2 hours. Some samples required up to 4 hours of immersion for a complete removal of organic matter. The tests were rinsed 3 times with reagent grade water (grade I) and dried under a laminar flow hood. In order to create subsamples for each sampled core depth, 7 – 12 tests were picked (minimum 7 tests for *C. neoteretis*, and

minimum 10 for *N. pachyderma*) and placed in micro-reaction hydrolysis vials pre-sterilised by heating at 550 °C. The number of subsamples within a sample was constrained by the number of available tests, and their level of preservation in a particular sample. To dissolve the tests, 8 μl of 6M HCl was added to the vials, and the solution was sealed under $N_2$ gas. The subsamples were hydrolysed at 110 °C for 6 hours in order to convert the original protein present in the tests into free amino acids. Following hydrolysis, the subsamples were evaporated in a vacuum desiccator. 4 μl of 0.01

M HCl with 10 μM L-*homo*arginine spike was added to the evaporate, and the subsamples were then injected onto a high-performance liquid chromatograph (HPLC). All measurements were performed at the Amino Acid Geochronology Laboratory, Northern Arizona University. The HPLC instrumentation and procedure used are described by Kaufman and Manley (1998). Inter-laboratory standards (Wehmiller, 1984; Table 2) were analysed to monitor instrument performance.


The extent of amino acid racemization was determined by analysing the peak-area ratio of D- and L- enantiomers (9 amino acids measured in total). Data analysis focused on Asp and Glu, as these amino acids are abundant in foraminifera protein and among the best resolved chromatographically (e.g. Kaufman et al., 2013), and they provide the basis of previous calibrated age equations.


### 3 Results

A total of 32 foraminifera samples (10 from core PS92/39-2, 13 from core PS92/45-1 and 9 from core PS92/54-1) subdivided into 241 subsamples (average: 7.5 subsamples per sample) were analysed. The average number of

foraminifera tests per subsample was 10.2 for *N. pachyderma* and 7.5 for *C. neoteretis* (Table 2). The results of amino acid analysis of all subsamples are contained in Supplementary Table S1.

In order to assess whether subsample size had a significant influence on the resulting D/L values, a further 47 subsamples of the foraminifera *N. pachyderma* were prepared, each with 20 individual tests. The results of this analysis





were then compared with 42 subsamples that comprised 10 tests. No statistically significant difference could be observed at the 0.01 probability level (the details of this analysis are contained in the Supplementary Information).

Outliers were removed by following the screening process of Kosnik and Kaufman (2008):

1. Subsamples with L-Ser / L-Asp values greater than 0.8 were excluded as this could be an indicator of contamination by modern amino acids.

2. The positive covariance of Asp and Glu acid D/L values in foraminifera is well known (e.g. Hearty et al., 2004; Kosnik and Kaufman, 2008); subsamples that deviated from the expected trend were excluded (Figure 3).

3. Remaining subsamples with D/L Asp or D/L Glu values outside the 2σ of the sample mean were rejected to emphasize the central tendency of the data.

This screening process resulted in the rejection of 68 subsamples, equivalent to 28.2 % of all subsamples. The overall rejection rate is higher than for most other studies as summarised by Kaufman et al. (2013), although in this study,
subsamples from core 39-2 account for over half (53%) of all rejected subsamples, despite that only 35 % of all subsamples was taken from this core. Only 18.8 % – typical of previous studies – of the subsamples from core 45-2 were rejected, and this core provided 40 % of all subsamples.

Asp and Glu D/L values show an increasing trend with depth for both foraminifera species (Figure 4) in all cores. D/L
values of Asp and Glu in core PS92/39-2 are lower at equivalent depths below seafloor than compared to PS92/45-2 and PS92/54-1 when samples from below 3 m are considered. Eight of the 32 samples exhibit mean Asp D/L values in reverse stratigraphic order (highlighted with bold in Table 2). Six of the reversed values overlap within 2σ errors with the sample from shallower depths, and only two mean Asp D/L values from *C. neoteretis* samples in core PS92/45-2 (at 4.45 m), and in core PS92/54-1 (at 1.79 m), lack such an overlap. Mean D/L Glu values also exhibit
stratigraphically reversed values in these samples, with overlaps within the error range of the overlying sample.

Mean D/L values of Asp and Glu are generally higher for *N. pachyderma* than for *C. neoteretis* for samples from the same depth (Table 2). This difference can be observed in both cores PS92/45-2 and PS92/54-1. It is difficult to quantify the differences in racemization rates between the two species, as little data is available for an extensive comparison, and some stratigraphic levels suffer from reversed D/L values. However, as a basic approximation, a plot of *N.*
*pachyderma* versus *C. neoteretis* mean D/L values implies that Asp racemizes approximately 17% faster in *N. pachyderma* than in *C. neoteretis*, while Glu racemizes about 23-26 % faster in *N. pachyderma* (Figure 5).

## 4 Discussion


### 4.1 Relation between D/L values and depth, and inter-specific differences

The extent of racemization generally increases with depth in both *N. pachyderma* and *C. neoteretis* samples and this conforms to the expected diagenetic behaviour of amino acids in foraminifera. Eight samples have stratigraphically





reversed mean D/L Asp and Glu values, but six of these values overlap within the 2σ uncertainty envelope with the overlying sample.

For a given sediment depth, the extent of racemization in *N. pachyderma* samples is lower in PS92/39-2 below 3 m than in the other two cores (Figure 4). This is consistent with the higher sedimentation rates in PS92/39-2 below 3 m,

compared to the other two cores, based on the independent age control of Wiers et al. (2019).

Differences in the rates of AAR (16% for Asp and 23-26% for Glu – Figure 5) between *N. pachyderma* and *C. neoteretis* tests from the same depths are similar to those documented in other AAR studies (e.g. King and Neville, 1977; Kaufman et al., 2013). Little is known about the extent and causes of the differences in racemization rates in

these two species, but differences of similar magnitude between D/L values of different taxa were previously reported in the literature. For example, Kaufman et al. (2013) found that Asp racemized 12-16 % faster in *Pulleniatina obliquiloculata* than in *N. pachyderma*. As *N. pachyderma* and *C. neoteretis* are amongst the most commonly occurring planktic and benthic foraminifera species in the Arctic, the quantification of the relative rates of racemization between the two species will aid future AAR studies in the Arctic and could be augmented by future laboratory heating

experiments.

Stratigraphically reversed D/L Asp values, which do not overlap within the 2σ uncertainty, only appear in *C. neoteretis* samples. One such reversal is in core PS92/45-2, in sample UAL17321 from 4.45 m core depth (total core length: 5.20 m). In this sample the mean D/L Asp value of 0.277 (σ = 0.022) is essentially equal to the mean value of 0.274 (σ =

0.021) from 3.12 m depth. It is difficult to explain the origin of the stratigraphically reversed mean values in the *C. neoteretis* samples. Sediment mixing offers one possible explanation, and may account for reversals on decimetre scales, but it seems less likely that metre-scale downward mixing – which would be required to explain the reversal discussed in PS92/45-2 – occurs. The small number (7) of foraminifera tests used in subsamples of *C. neoteretis* could make the D/L values more susceptible to variations in individual tests. For example, Lougheed et al. (2018) recently

highlighted the large heterogeneity in the age distribution of foraminifera obtained from discrete depth intervals using [14]C dating of single foraminifera.

### 4.2 Relation between D/L values and sample age

A biplot of D/L values against sample ages reveals that, as expected, the rate of Asp and Glu racemization is higher in younger samples and decreases with sample age as the reaction progresses towards equilibrium (Figure 6). D/L values increase in a predictable manner over time as shown by least square regression power curves fitted to the mean D/L Asp and Glu values for each of the cores. An exceptionally good fit for *N. pachyderma* samples is achieved in core PS92/45-2 (with $R^2$ values of 0.99 and 0.98 for D/L Asp and Glu, respectively; for individual regression lines see

Supplementary Figure 1). In *C. neoteretis* samples, both Asp and Glu appear to racemize at similar rates in cores PS92/45-2 and PS92/54-1. The dissimilar AAR rates between samples of comparable ages from different cores may originate from differences in sedimentation rates between the cores, post-depositional environmental factors, including differences in geothermal gradients or variable diagenetic processes.



### 4.3 Paleotemperature and other possible effects

In monospecific foraminifera samples, the rate of racemization is primarily controlled by the integrated post-depositional temperature (Kaufman et al., 2013). Therefore, if the D/L value and age of a sample are known, the AAR age equation can be solved for paleotemperatures. Here we make a rough estimation of the difference in the effective diagenetic temperatures required to account for the offset between known-age-equivalent D/L values between the coring sites. To achieve this, we relied on the temperature equation originally derived by Kaufman (2006) using simple power law kinetics for Asp in the foraminifera species, *Pulleniatina obliquiloculata,* and increased the D/L values of our *N. pachyderma* samples by 16% to account for the lower rate of racemization in this taxon, as established by Kaufman et al. (2013). For comparison between the three core sites, we determined the approximate D/L Asp values predicted by power-fit functions for each core at 150 ka, then the effective diagenetic temperatures integrated over the past 150 ka were derived from the temperature equation. The results imply that temperature differences ($\Delta$T) of ~ 1.6 – 4.0 ºC are required to account for the observed differences in racemization rates between the cores over the past 150 ka (Table 3). Details on paleotemperature calculations are contained in the Supplementary Information.

The estimated difference in effective diagenetic temperatures between PS92/39-2 and PS92/54-1 (4 ± 1.6 ºC) cannot be easily accounted for by changes in bottom water temperatures. Modern bottom water temperatures at the 3 coring locations are relatively similar (between – 0.25 and – 0.76 °C) (Table 1), but we do not know how similar these remained across glacial cycles.

An alternate explanation for this temperature difference could be differences in the geothermal gradient among sites. Unfortunately, no direct measurements of heat flow were obtained at the coring stations during TRANSSIZ 2015 and existing heat flow data from the Yermak Plateau are relatively sparse (Okay and Crane, 1993; Shephard et al., 2018). Existing measurements do indicate that regions of the Yermak Plateau are characterised by relatively high heat flow (>100 mW/m$^2$), but with considerable spatial variability. Most measurements lie in the range of 50 to 100 mW/m$^2$ (Shephard et al., 2018). If we use a thermal conductivity of 1.17 W/mK, reported from measurements on a core obtained from the Yermak Plateau (Shephard et al., 2018), heat flow of 50-100 m/Wm$^2$ translates into geothermal gradients of 43 to 85 ºC/km (0.043 to 0.085 ºC/m). At a depth of 6 m below the seafloor, this range of geothermal gradients would amount to a present-day in-situ temperature difference of only 0.25 ºC. Thus, it seems unlikely that local heat flow variations would significantly affect rates of racemization at the shallow sediment depth of these samples.

Aside from in-situ temperatures, Sejrup and Haugen (1994) theorised that variations in post-depositional microbial activity could have a profound impact on AAR, and that the extent of microbial activity is related to sedimentation rates. Shells deposited in settings with low sedimentation rates could have been exposed longer to microbial activity at the seabed than shells that were buried rapidly, and therefore more rapidly removed from the taphonomically active zone. Such increased exposure could result in a higher degree of protein and amino acid degradation in the early diagenetic phase, giving rise to higher initial D/L values. The sedimentation rate in core PS92/39-2 is estimated to be around 4.5 cm/ka during MIS 5 (assigned to the depth interval between 3.5 – 6 m), two and a half times that of the





sedimentation rate at the other two core sites (Table 3). The comparatively lower D/L values in core PS92/39-2 could reflect the faster burial and thereby better preservation of indigenous proteins in tests from the 3.5–6 m depth interval,

yet the exact mechanisms of microbially mediated racemization remain unresolved. Currently we lack sufficient data to more fully explore the reasons for inter-site offsets in the AAR rates.

### 4.4 Comparing AAR age models

In order to assess the validity of the previously derived Arctic-specific (Kaufman et al., 2008) and globally calibrated (Kaufman et al., 2013) age equations at the Yermak Plateau, ages predicted by the two models were generated using the mean D/L Asp and Glu values from *N. pachyderma* and *C. neoteretis* tests in this study (Table 4). No ages using the Arctic-specific age equation were generated for *C. neoteretis* samples, as this calibration is based on AAR in *N. pachyderma* tests only. Ages predicted by the models were plotted against the corresponding D/L values and displayed

for reference in the biplot of D/L values against sample age (Figure 6). Age uncertainties were derived using the intra-sample variability (± 1 σ) propagated through the age equations.

Modelled ages generated using the globally calibrated age equation of Kaufman et al. (2013) and mean D/L Asp values in *N. pachyderma* are most similar to the sample ages of Wiers et al. (2019) (Figure 6). The Arctic-specific age equation

of Kaufman et al. (2008) results in significantly younger ages than those based on the Wiers et al. (2019) age model (Figure 6). This implies that the Arctic-specific age equation cannot be used to constrain the ages of sediments from the Yermak Plateau.

These results generally support the rates of AAR determined for other deep-sea (cold bottom water) sites, further

highlighting the peculiar high rate of racemization indicated by previous analyses of central Arctic sediments. For example, previously published (Kaufman et al., 2008) mean D/L Asp values for *N. pachyderma* samples from a central Arctic core from the Lomonosov Ridge (96/12-1PC) are around 0.4 for samples dated to 85 – 123 ka. The globally calibrated age equation predicts approximate ages of 263 ka for this D/L Asp value (with ± 15.8 ka error, assuming an uncertainty of 6%, typical of mean D/L values in this study), and samples with similar D/L values from the Yermak

Plateau would date to around 232 ± 14 ka. The considerable differences could indicate different post-depositional temperatures, diagenetic processes, issues with sample handling, storage, and processing or may point towards errors in the existing age models of central Arctic sediments. Further studies are clearly required to investigate these options.

AAR ages based on D/L Asp values and the globally calibrated AAR curve strongly correlate with the independent

ages of Wiers et al. (2019) for both species (Figure 7); however, there is a significant scatter in AAR ages older than ~130 ka. Excluding all samples with stratigraphically reversed D/L Asp values (regardless whether there is an overlap within 2σ errors or not) increases the correlation coefficient (r) to 0.96 in *N. pachyderma*, and to 0.99 in *C. neoteretis* samples. While well-correlated, the globally calibrated age equation slightly (~on average 2%) underestimates the ages of *N. pachyderma* samples, with the highest level of agreement between the ages observed in core PS92/45-2. The

ages of *C. neoteretis* samples are mostly underestimated – on average by 32%. This is expected considering the lower racemization rates of *C. neoteretis* observed in this study (approximately 16% slower than *N. pachyderma*). Both sets of data are coupled with high values of scatter, with differences between the modelled ages and the ages based on



Wiers et al. (2019) reaching over 50 % of the sample age in some cases. Nevertheless, the globally calibrated age equation provides a reliable age approximation for our Arctic Ocean sediments, when D/L Asp values from *N. pachyderma* samples are utilised.


## 5   Conclusions

- The extent of racemization of Asp and Glu increases with increasing age in both *N. pachyderma* and *C. neoteretis* samples and is higher in *N. pachyderma* than in *C. neoteretis* from the same stratigraphic levels.


- The globally calibrated age equation of Kaufman et al. (2013) provides a reliable age control for sediments from the Yermak Plateau. The Arctic-specific calibrated age equation of Kaufman et al. (2008) does not support the independent age-depth model in this area, and is not applicable to sediments from the Yermak Plateau.

- Ages calculated with the globally calibrated age equation of Kaufman et al. (2013) show the highest level of agreement with sample ages when D/L Asp values from *N. pachyderma* samples are used. Ages obtained using the globally calibrated age equation and D/L Asp values in *C. neoteretis* samples consistently appear younger (on average by 32%) than the independent ages, as expected given the lower rate of AAR in this species.


- The results highlight the need for further studies to test and explain the origin of the higher racemization rates in foraminifera reported in previous studies from central Arctic sediments.


## Data availability

The full suite of data – including concentration data for all amino acids analysed – can be obtained from the authors and will be accessible via the Bolin Centre for Climate Research database.


## Author contribution

GW analysed the data and prepared the manuscript with contributions from all authors. DK contributed interpretation of AAR results, FM provided age model tuning, MF, JM and JW contributed material, chronological and oceanographic interpretation, MO coordinated the study and contributed palaeoceanographic interpretation.


## Competing interests

The authors declare that they have no conflict of interest.


## Acknowledgements

This study was funded by the Swedish Research Council (VR) (Grant DNR-2016-05092) and the National Science Foundation. We thank the Captain and the crew of RV Polarstern, and the participants of the TRANSSIZ 2015 Expedition for facilitating data collection during PS92. Katherine Whitacre provided laboratory support at the Amino Acid Geochronology Laboratory, Northern Arizona University.




**Appendix A**

Prior to alignment, the kARM/k timeseries and the $\delta^{18}O_{LR}$ stack were rescaled between -1 and 1 to improve comparability. The alignment function is then inferred at any point on the sediment core depth scale by a linear interpolation between three proposed depth-age nodes–a starting node, an ending node, and a perturbed node at a random location between the starting and ending nodes. The nodes strictly follow depth-age paths that do not violate the principle of superposition in order to ensure a monotonic relationship between the depth of the sediment core and

the age of the $\delta^{18}O_{LR}$ stack. The algorithm determines the set of nodes that give a good alignment between the kARM/k timeseries and the $\delta^{18}O_{LR}$ stack as defined by a small residual standard deviation and a high coefficient of correlation. As an additional condition, we require that the gradient of the depth-age alignment function should be reasonably smooth, hence assuming that at our coring sites sedimentation rates have not undergone excessively rapid shifts or large fluctuations.


The requirement of a good match between the kARM/k records and the $\delta^{18}O_{LR}$ stack, and a smooth alignment function is here devised in a Bayesian formulation sensitive to specification of a conjugate prior distribution and a conjugate likelihood distribution of the alignment function. The priors specify *i*) the age distribution of the first and last kARM/k values on the $\delta^{18}O_{LR}$ age scale (here set as non-informative uniform ranging 10 kyrs), and *ii*) the degree of

autocorrelation of the sedimentation rates (here set as truncated uniform between -50% and +50%). The latter dictates how much the accumulation rate of a particular depth depends on the depth above it, which in turn controls the gradient of the depth-age alignment function. The likelihood distribution of the alignment function weighs the competing needs of small standard deviations of the data residuals and a high correlation between the kARM/k and the $\delta^{18}O_{LR}$ values. Finally, a conditional posterior distribution of alignment functions proportional to the product of prior and likelihood

is inferred. Calculation of the posterior proceeds by sampling an initial value for each unknown parameter from the associated prior distributions using a reversible jump Markov chain Monte Carlo (MCMC) sampling (Vihola, 2012). In the alignment problem posed here, the automated MCMC method consists of the following steps. Starting from an initial age for the first and last kARM/k values on the $\delta^{18}O_{LR}$ timescale that defines an initial match between the kARM/k and $\delta^{18}O_{LR}$ timeseries, and by setting an initial value for the rate of sedimentation, it continues by:


1.   Proposing a new "candidate" alignment function by adding a new depth-age node at a random location between the starting and ending nodes along the depth scale of the sediment core.

2.   Accepting or rejecting the candidate alignment function according to its posterior probability using the
385       Metropolis-Hasting algorithm (Metropolis, 1953; Hastings, 1970), whereby the posterior probability is higher for alignment functions that yield a closer match (i.e. a smaller residual standard deviation and a high coefficient of correlation).

3.   Repeat from step 1 for 106 iterations.






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





**Figure captions**

**Figure 1. Locations of the studied cores and water depth at core sites. Basemap: IBCAO, Jakobsson et al. (2012).**


**Figure 2. The age-depth models (Wiers et al., 2019) of the cores in this study are based on correlation of environmental magnetic parameters (kARM/k) and the global benthic δ¹⁸O stack of Lisiecki and Raymo (2005).**

**Figure 3. Covariance of aspartic acid and glutamic acid D/L values in all subsamples. Open symbols indicate rejected results,**
**with the rejection criterion indicated by the symbol colour.**

**Figure 4. Extent of racemization (mean D/L in aspartic acid and glutamic acid) in samples of *N. pachyderma* and *C. neoteretis* plotted against depth in sediment cores PS92/39-2, PS92/45-2 and PS92/54-1. Error bars represent ±1σ deviation from the sample mean. Least square regression lines (power fit) are shown for all cores. Data listed in Table 2.**


**Figure 5. Extent of racemization (D/L) for aspartic and glutamic acids in *N. pachyderma* versus *C. neoteretis* samples from cores PS92/45-2 and PS92/54-1, with linear regression and line of equality.**

**Figure 6. Mean D/L values of aspartic and glutamic acids plotted against the independent ages of Wiers et al. (2019) for *N.***
***pachyderma* and *C. neoteretis* samples from cores PS92/39-2, PS92/45-2 and PS92/54-1. Ages predicted (based on D/L Asp and Glu values) by the globally calibrated age equation of Kaufman et al. (2013) and the Arctic-specific age equation are also displayed for reference. Error bars represent ±1σ deviation from the mean D/L value of the sample, and age uncertainty.**

**Figure 7. Comparison of sample ages for the two foraminifera taxa as predicted by the globally calibrated age equation of**
**Kaufman et al. (2013) based on D/L Asp values and ages assigned to the samples based on the age model developed by Wiers et al. (2019), with line of equality.**



Figure 1.

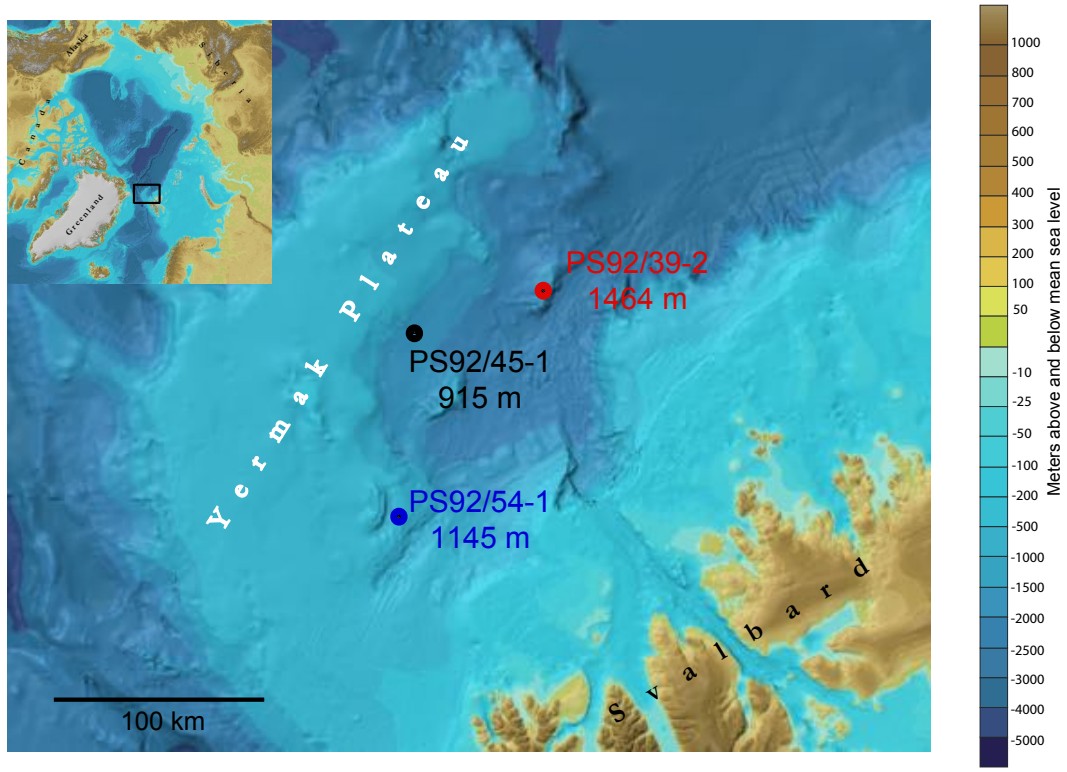



Figure 2





Figure 3

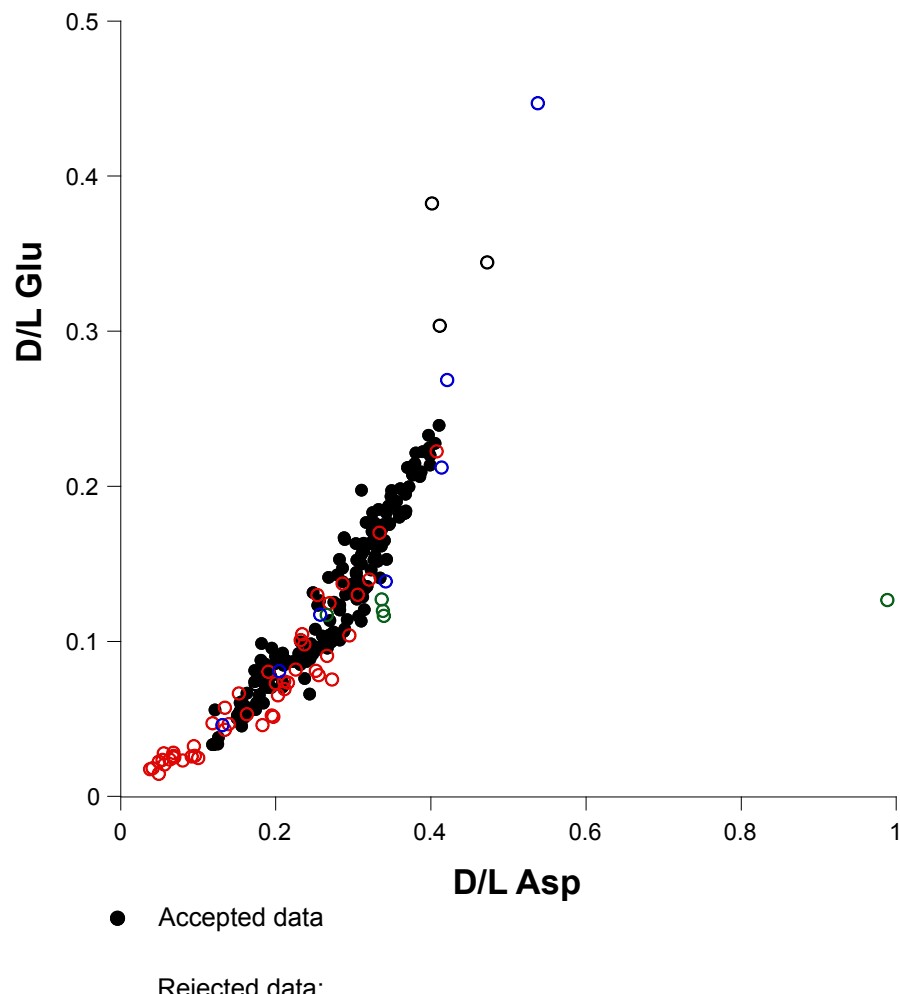





Figure 4

*Neogloboquadrina pachyderma*

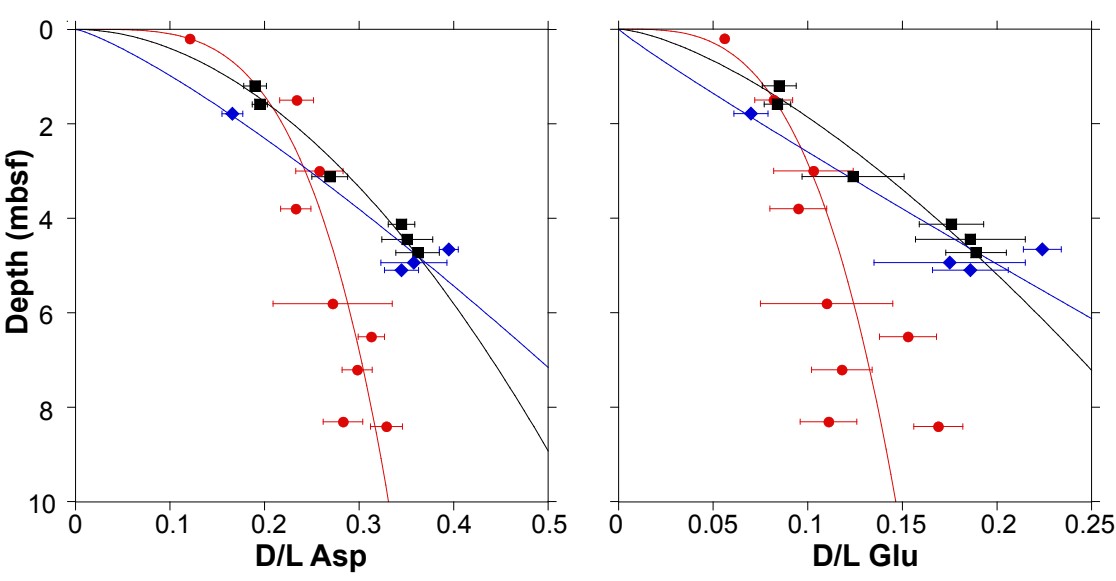

● PS92/39-2
■ PS92/45-2
◆ PS92/54-1

*Cassidulina neoteretis*

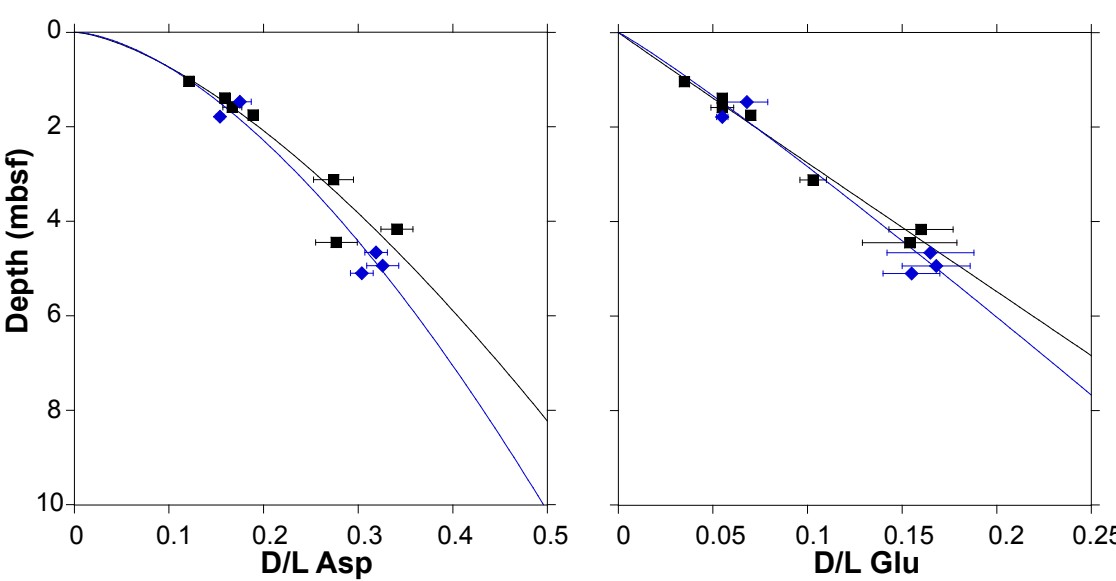





Figure 5

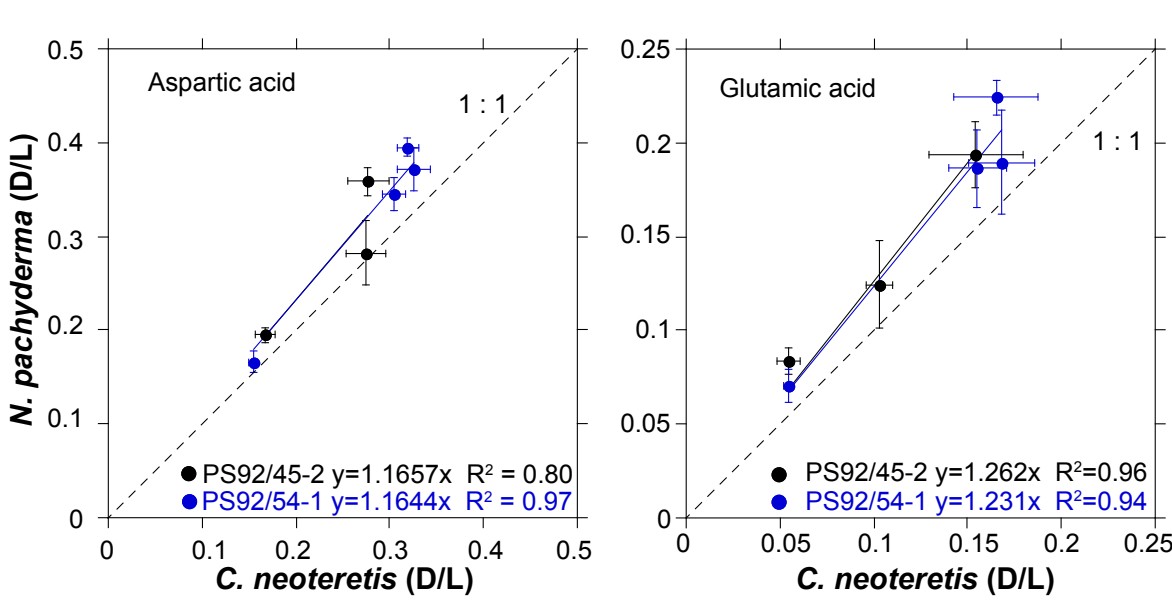





Figure 6

### *Neogloboquadrina pachyderma*

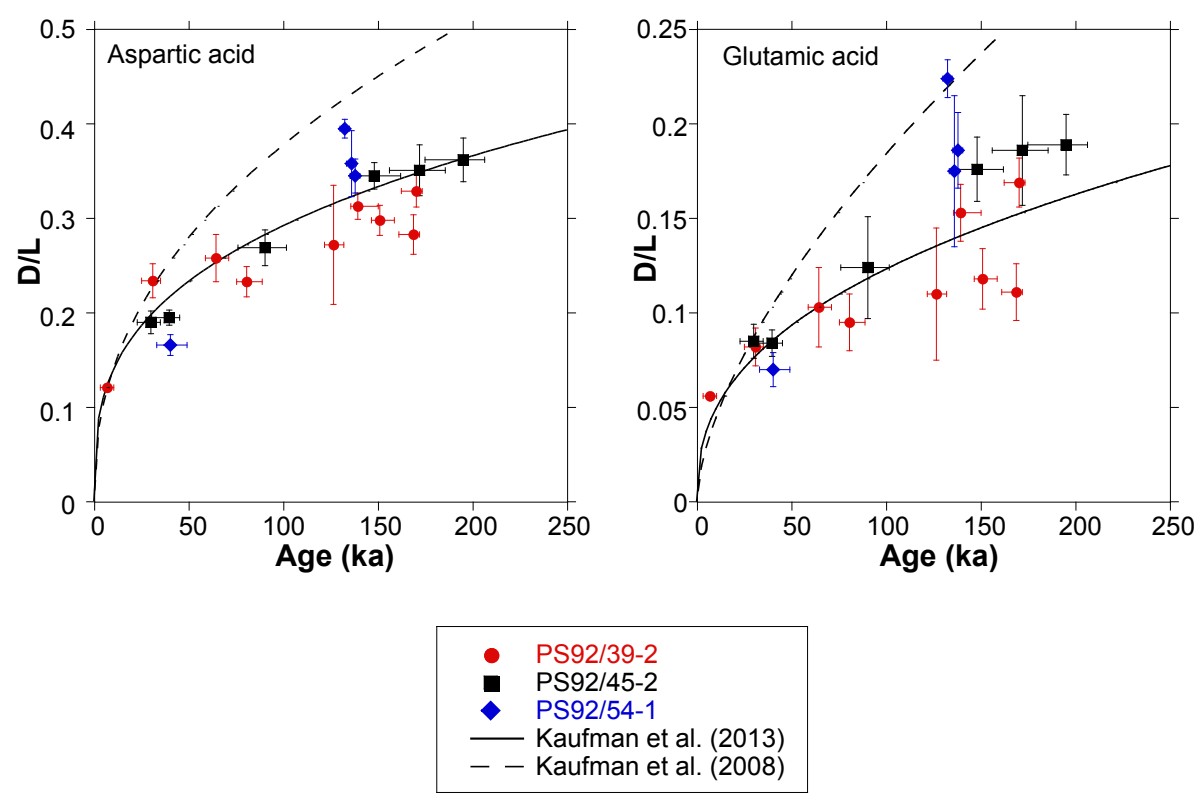

### *Cassidulina neoteretis*

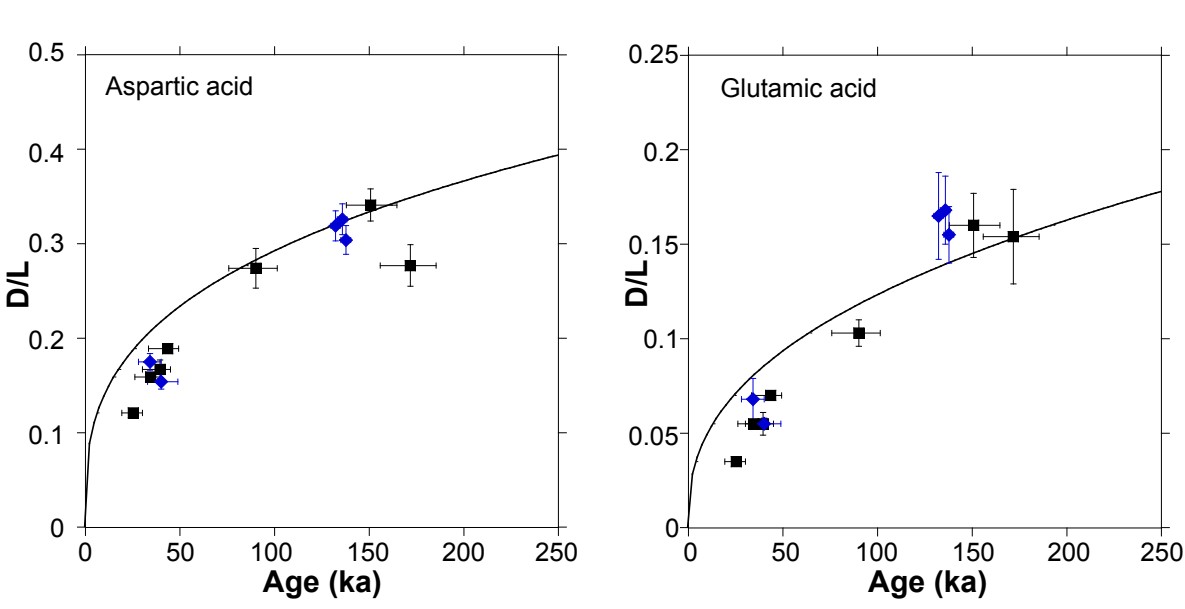





Figure 7

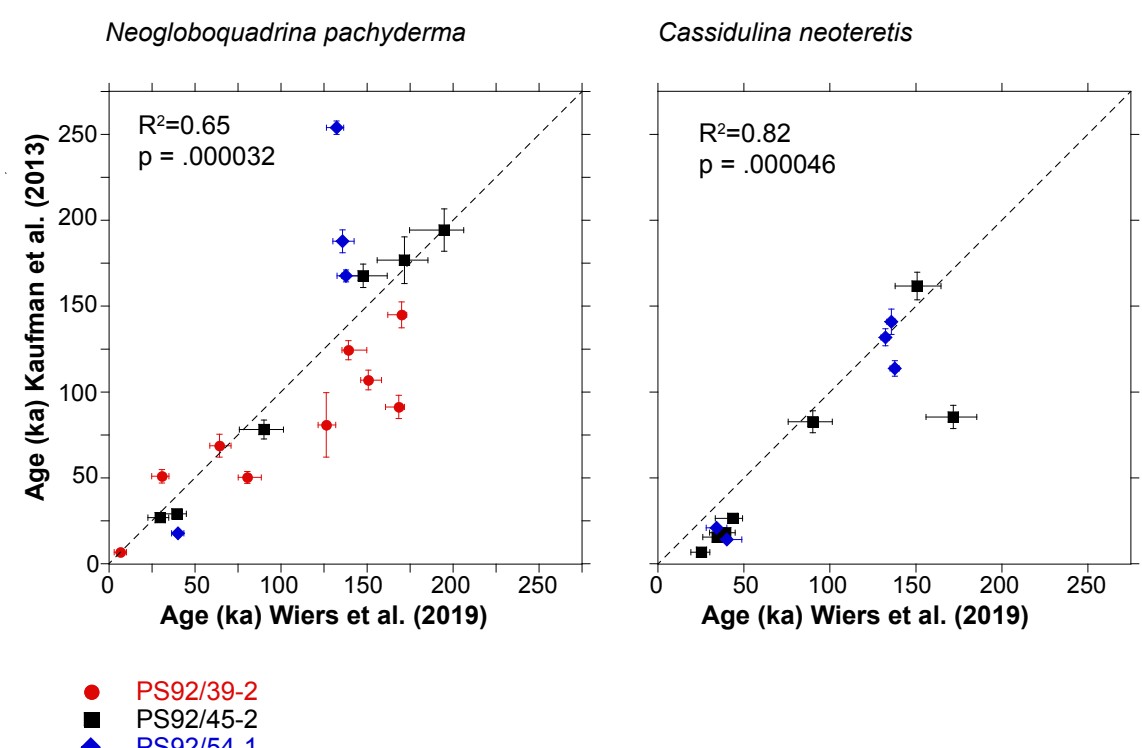





**Tables**


**Table 1. Sediment cores analysed in this study.**

| Core | Core type | Latitude (°N) | Longitude (°E) | Water depth (m) | Core length (m) | Bottom water temperature (°C) |
|------|-----------|---------------|----------------|-----------------|-----------------|-------------------------------|
| PS92/39-2 | kastenlot core | 81.94983 | 13.82833 | 1464 | 8.600 | - 0.76 |
| PS92/45-2 | gravity core | 81.89333 | 9.768833 | 915 | 5.195 | - 0.25 |
| PS92/54-1 | gravity core | 81.123 | 8.466167 | 1145 | 4.145 | - 0.43 |

**Table 2. Summary of Aspartic acid and glutamic acid mean D/L values (following data screening) and independent ages (based on Wiers et al. (2019) of samples analysed in this study. n = number of subsamples analysed; ex = number of excluded**
**subsamples. Cells highlighted in yellow/green indicate core depths sampled for both Neogloboquadrina pachyderma and Cassidulina neoteretis. Values in bold are mean D/L values that show depth reversals. Interlaboratory standards (ILC A, B, C) analysed during the course of this study are also shown.**

| UAL ID | Core | Depth (mbsf) | Age (ka) | n | ex | Number of tests | D/L Asp | σ | D/L Glu | σ |
|--------|------|--------------|----------|---|----|-----------------|---------|---|---------|---|
| *Neogloboquadrina pachyderma* | | | | | | | | | | |
| 15544 | PS92/39-2 | 0.20 | 6.7 | 7 | 6 | 10 | 0.121 | - | 0.056 | - |
| 15545 | PS92/39-2 | 0.35 | 8.9 | 4 | 4 | 10 | - | - | - | - |
| 15546 | PS92/39-2 | 1.50 | 30.8 | 7 | 3 | 10 | 0.234 | 0.018 | 0.082 | 0.010 |
| 15547 | PS92/39-2 | 3.00 | 64.2 | 8 | 0 | 10 | 0.258 | 0.025 | 0.103 | 0.021 |
| 15548 | PS92/39-2 | 3.80 | 80.4 | 10 | 2 | 10 | **0.233** | 0.016 | **0.095** | 0.015 |
| 15549 | PS92/39-2 | 5.81 | 126.3 | 9 | 5 | 10 | 0.272 | 0.063 | 0.110 | 0.035 |
| 15550 | PS92/39-2 | 6.51 | 139.2 | 9 | 1 | 10 | 0.313 | 0.014 | 0.153 | 0.015 |
| 15551 | PS92/39-2 | 7.21 | 150.8 | 10 | 6 | 10 | **0.298** | 0.016 | **0.118** | 0.016 |
| 15552 | PS92/39-2 | 8.31 | 168.5 | 10 | 5 | 10 | **0.283** | 0.021 | **0.111** | 0.015 |
| 15553 | PS92/39-2 | 8.41 | 170.1 | 10 | 4 | 10 | 0.329 | 0.017 | 0.169 | 0.013 |
| 16814 | PS92/45-2 | 1.20 | 29.8 | 9 | 3 | 10 | 0.190 | 0.012 | 0.085 | 0.009 |
| 16815 | PS92/45-2 | 1.59 | 39.5 | 11 | 0 | 10 | 0.195 | 0.008 | 0.084 | 0.007 |
| 16817 | PS92/45-2 | 3.12 | 90.1 | 8 | 4 | 10 | 0.269 | 0.019 | 0.124 | 0.027 |
| 16818 | PS92/45-2 | 4.13 | 147.8 | 7 | 3 | 10 | 0.345 | 0.014 | 0.176 | 0.017 |
| 17531 | PS92/45-2 | 4.45 | 171.8 | 11 | 0 | 12 | 0.351 | 0.027 | 0.186 | 0.029 |
| 16819 | PS92/45-2 | 4.73 | 194.9 | 6 | 2 | 10 | 0.362 | 0.023 | 0.189 | 0.016 |
| 16822 | PS92/54-1 | 1.79 | 40.1 | 7 | 1 | 10 | 0.166 | 0.011 | 0.070 | 0.009 |
| 16826 | PS92/54-1 | 4.66 | 132.3 | 15 | 7 | 12 | 0.395 | 0.010 | 0.224 | 0.010 |
| 16827 | PS92/54-1 | 4.94 | 135.8 | 6 | 1 | 10 | **0.358** | 0.035 | **0.175** | 0.040 |
| 16828 | PS92/54-1 | 5.1 | 137.8 | 4 | 1 | 10 | **0.345** | 0.018 | **0.186** | 0.020 |
| *Cassidulina neoteretis* | | | | | | | | | | |
| 17315 | PS92/45-2 | 1.04 | 25.4 | 7 | 1 | 7 | 0.121 | 0.003 | 0.035 | 0.002 |
| 17316 | PS92/45-2 | 1.39 | 34.5 | 2 | 1 | 7 | 0.159 | - | 0.055 | - |
| 17317 | PS92/45-2 | 1.59 | 39.5 | 7 | 1 | 7 | 0.167 | 0.010 | 0.055 | 0.006 |





| 17318 | PS92/45-2 | 1.75 | 43.5 | 3 | 2 | 7 | 0.189 | - | 0.070 | - |
| 17319 | PS92/45-2 | 3.12 | 90.1 | 8 | 2 | 10 | 0.274 | 0.021 | 0.103 | 0.007 |
| 17320 | PS92/45-2 | 4.17 | 150.8 | 7 | 3 | 7 | 0.341 | 0.017 | 0.160 | 0.017 |
| 17321 | PS92/45-2 | 4.45 | 171.8 | 10 | 3 | 10 | **0.277** | 0.022 | **0.154** | 0.025 |
| 17322 | PS92/54-1 | 1.47 | 34.2 | 5 | 0 | 7 | 0.175 | 0.012 | 0.068 | 0.011 |
| 17323 | PS92/54-1 | 1.79 | 40.1 | 6 | 0 | 7 | **0.154** | 0.004 | **0.055** | 0.003 |
| 17324 | PS92/54-1 | 4.66 | 132.3 | 7 | 2 | 7 | 0.319 | 0.012 | 0.165 | 0.023 |
| 17325 | PS92/54-1 | 4.94 | 135.8 | 6 | 2 | 7 | 0.326 | 0.017 | 0.168 | 0.018 |
| 17326 | PS92/54-1 | 5.1 | 137.8 | 5 | 0 | 7 | **0.304** | 0.012 | **0.155** | 0.015 |
| *Interlaboratory standards* | | | | | | | | | | |
| ILC-A | N/A | N/A | N/A | N/A | N/A | N/A | 0.412 | | 0.225 | |
| ILC-B | N/A | N/A | N/A | N/A | N/A | N/A | 0.718 | | 0.453 | |
| ILC-C | N/A | N/A | N/A | N/A | N/A | N/A | 0.842 | | 0.845 | |





**Table 3.** Approximate effective diagenetic temperatures at the three core sites integrated over the past 150 ka.

| Core | Estimated effective diagenetic temperatures over the past 150 ka | Approximate average sedimentation rate during MIS 5 (cm/ka) |
|---|---|---|
| PS92/39-2 | + 4.2 ± 0.8 | 4.5 |
| PS92/45-2 | + 5.8 ± 0.8 | 2 |
| PS92/54-1 | + 8.2 ± 0.8 | 1.9 |


**Table 4.** Independent sample ages based on Wiers et al. (2019), and sample ages and uncertainties based on the globally calibrated age equation of Kaufman et al. (2013) and the Arctic-specific age equation of Kaufman et al. (2008) based on mean D/L values of Asp and Glu.

| UAL ID | Core | Depth (mbsf) | Age (ka) based on Wiers et al. (2019) | Age Asp Kaufman et al. 2013 (ka) | | | Age Kaufman et al. 2008 (ka) | Asp | | Age Glu Kaufman et al. 2013 (ka) | | | Age Glu Kaufman et al. 2008 (ka) | | |
|---|---|---|---|---|---|---|---|---|---|---|---|---|---|---|---|
| *Neogloboquadrina pachyderma* | | | | | | | | | | | | | | | |
| 15544 | PS92/39-2 | 0.20 | 6.7 | 6.7 | ± | | 7.3 | | | 14.2 | | | 15.0 | | |
| 15546 | PS92/39-2 | 1.50 | 30.8 | 51.0 | ± | 3.9 | 33.4 | ± | 2.6 | 36.7 | ± | 4.5 | 27.5 | ± | 3.4 |
| 15547 | PS92/39-2 | 3.00 | 64.2 | 68.8 | ± | 6.7 | 41.9 | ± | 4.1 | 64.7 | ± | 13.2 | 39.7 | ± | 8.1 |
| 15548 | PS92/39-2 | 3.80 | 80.4 | 50.3 | ± | 3.5 | 33.1 | ± | 2.3 | 50.2 | ± | 3.2 | 34.8 | ± | 5.5 |
| 15549 | PS92/39-2 | 5.81 | 126.3 | 87.4 | ± | 18.7 | 50.1 | ± | 10.9 | 83.2 | ± | 29.9 | 44.1 | ± | 14.0 |
| 15550 | PS92/39-2 | 6.51 | 139.2 | 128.1 | ± | 5.6 | 66.7 | ± | 2.9 | 161.8 | ± | 19.5 | 74.7 | ± | 7.3 |
| 15551 | PS92/39-2 | 7.21 | 150.8 | 112.6 | ± | 5.7 | 60.6 | ± | 3.1 | 102.5 | ± | 11.6 | 49.3 | ± | 6.7 |
| 15552 | PS92/39-2 | 8.31 | 168.5 | 96.4 | ± | 6.8 | 53.9 | ± | 3.8 | 83.2 | ± | 10.9 | 44.7 | ± | 6.0 |
| 15553 | PS92/39-2 | 8.41 | 170.1 | 146.3 | ± | 7.5 | 73.7 | ± | 3.8 | 199.2 | ± | 27.1 | 87.6 | ± | 6.7 |
| 16814 | PS92/45-2 | 1.20 | 29.8 | 26.9 | ± | 1.7 | 20.7 | ± | 1.3 | 40.1 | ± | 4.2 | 29.2 | ± | 3.1 |
| 16815 | PS92/45-2 | 1.59 | 39.5 | 29.2 | ± | 1.2 | 22.0 | ± | 0.9 | 39.0 | ± | 3.2 | 28.6 | ± | 2.4 |
| 16817 | PS92/45-2 | 3.12 | 90.1 | 90.3 | ± | 5.5 | 51.4 | ± | 3.3 | 104.6 | ± | 19.2 | 53.4 | ± | 11.6 |
| 16818 | PS92/45-2 | 4.13 | 147.8 | 167.6 | ± | 6.8 | 81.7 | ± | 3.3 | 244.7 | ± | 23.6 | 93.4 | ± | 9.0 |
| 17531 | PS92/45-2 | 4.45 | 171.8 | 186.2 | ± | 13.6 | 88.3 | ± | 6.5 | 303.8 | ± | 38.0 | 102.1 | ± | 15.9 |
| 16819 | PS92/45-2 | 4.73 | 194.9 | 194.3 | ± | 12.3 | 91.2 | ± | 5.8 | 292.1 | ± | 24.7 | 104.7 | ± | 8.9 |
| 16822 | PS92/54-1 | 1.79 | 40.1 | 18.1 | ± | 1.2 | 15.4 | ± | 1.0 | 25.7 | ± | 3.6 | 21.4 | ± | 2.7 |
| 16826 | PS92/54-1 | 4.66 | 132.3 | 253.9 | ± | 6.4 | 111.5 | ± | 2.8 | 445.6 | ± | 19.9 | 137.4 | ± | 6.1 |
| 16827 | PS92/54-1 | 4.94 | 135.8 | 209.5 | ± | 18.4 | 96.5 | ± | 8.7 | 296.0 | ± | 43.6 | 92.6 | ± | 21.2 |
| 16828 | PS92/54-1 | 5.10 | 137.8 | 167.6 | ± | 8.7 | 81.7 | ± | 4.3 | 280.8 | ± | 30.2 | 102.1 | ± | 11.0 |
| *Cassidulina neoteretis* | | | | | | | | | | | | | | | |
| 17315 | PS92/45-2 | 1.04 | 25.4 | 6.7 | ± | 0.2 | | | | 4.4 | ± | 0.3 | | | |
| 17316 | PS92/45-2 | 1.39 | 34.5 | 15.6 | ± | | | | | 13.6 | ± | - | | | |
| 17317 | PS92/45-2 | 1.59 | 39.5 | 18.1 | ± | 1.1 | | | | 13.6 | ± | 1.5 | | | |
| 17318 | PS92/45-2 | 1.75 | 43.5 | 26.5 | ± | | | | | 24.8 | ± | - | | | |
| 17319 | PS92/45-2 | 3.12 | 90.1 | 82.7 | ± | 6.3 | | | | 64.7 | ± | 4.4 | | | |
| 17320 | PS92/45-2 | 4.17 | 150.8 | 161.8 | ± | 8.1 | | | | 193.1 | ± | 20.5 | | | |
| 17321 | PS92/45-2 | 4.45 | 171.8 | 85.5 | ± | 6.8 | | | | 175.6 | ± | 28.5 | | | |
| 17322 | PS92/54-1 | 1.47 | 34.2 | 20.9 | ± | 1.4 | | | | 23.0 | ± | 3.7 | | | |
| 17323 | PS92/54-1 | 1.79 | 40.1 | 14.1 | ± | 0.4 | | | | 13.6 | ± | 0.7 | | | |

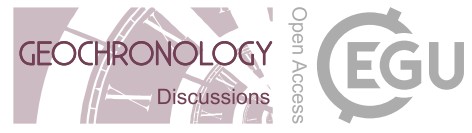

| 17324 | PS92/54-1 | 4.66 | 132.3 | 131.9 | ± | 5.0 | 208.5 | ± | 29.1 |
| 17325 | PS92/54-1 | 4.94 | 135.8 | 140.9 | ± | 7.3 | 218.0 | ± | 23.4 |
| 17326 | PS92/54-1 | 5.10 | 137.8 | 113.8 | ± | 4.5 | 178.5 | ± | 17.3 |
