# Peer review of "Amino acid racemization in Quaternary foraminifera from the Yermak Plateau"

_Geochronology, 2019_

## Referee Comment (RC1) · Anonymous Referee #1 · 20 Aug 2019

The study present amino acid data from three sediment cores raised from the eastern slope of the Yermak Plateau in the Arctic Ocean. Such studies have the possibility both to contribute in efforts to establish a reliable chronology and in efforts regarding identifying variability in bottom temperatures, both issues have been challenging in the Arctic ocean. The paper is well written and present an important set of data and deserve to be published. However, the discussion could be strengthen and the paper could be of larger general interest if the authors in a moderate to minor revision consider the following issues:

1.How good is the presented chronology beyond radiocarbon? Give some of the assumption it is based on and also the challenges in using a lithological/magnetic parameters in correlation in this geological setting. The age model for the older part of the

core with the longest record seems reasonable, however the actual correlation seems not very convincing? 2. IRD. In discussing ratios not in stratigraphic order, the authors should also discuss the possibility of reworking by sea ice and ice berg. Presentation of an iRD curve and some biostratigraphic data, if available, could be useful for this discussion. 3. Hydrography. The authors should present the main watermasses in the Arctic ocean basin and give the location of the cores relative to these. In the discussion of the diageneic temperatures it should befocused on possible changes in the deep water temperature and an evaluation of how this could influence on the amino acid diagenesis. 4. In the discussion of the wider implication of the data it should be a more focused on published data on the same species from i basins with possibly different temperature histories,and investigate of the amino cid data could contribute to a discussion of changing deep water temperatures related to the climatic variability and locus of deep water formation ( early attempt on this in Sejrup and Haugen 1992). 5. Background is generally good, however, it should be mentioned the large early work of D.L. Clark and others who in a number of papers suggested uniformly extremely low sedimentation rates in the Arctic ocean based mostly on interpretation of magnetic reversals. Amino acid data where actually the basis for some of the early challenging of this framework (Xuan and Channell 2010 intro ).

---

## Referee Comment (RC2) · Anonymous Referee #2 · 1 Sep 2019

The authors present an interesting study about the amino acid racemization extent in the planktic foraminifera Neogloboquadrina pachyderma and the benthic species Cassidulina neoteretis collected at some depths in 3 cores drilled in the Yermak Plateau, in the Artic Ocean. The results obtained here contribute to increase the knowledge of amino acid racemization in foraminifera species, especially for the establishment of a reliable chronological model for the Artic, and the processes that may affect D/L values. In my view, the this study is of general interest, and the data set is important. Therefore, it is suitable to be published in Geochronology after minor revision:

1.-In my view the authors should amplify the discussion regarding the influence of temperature and other factors in the D/L values that they observed. Planktic forams are subjected to marine currents and may remain in the water column for some time (hun-

dreds?). This may produce the accumulation of tests with different ages in the same layer (time-averaging) and that these tests may have been racemized at different rates depending on original location, the place of the water column, etc. In fact, the authors indicated that "Lougheed et al. (2018) recently highlighted the large heterogeneity in the age distribution of foraminifera obtained from discrete depth intervals using 14C dating of single foraminifera". Moreover, the cores were drilled at different positions with marked diverse depths. Also, it has to be considered that the Yerkman Plateau is located in an area with interaction between Artic and Atlantic waters. The dominant colder or warmer currents may have affected the racemization rates. Do they authors have information about temperature gradients or water currents in the area?

2.-The results reveal an good match with the age equations of Kaufman et al. (2013) calculated using diverse species from sites of the Artic, Atlantic and Pacific Oceans (with different environmental contidions, e.g. temperatures). However, the age model of Kaufman et al. (2008), calculated for N. pachyderma of the Artic Ocean was not applicable here. The authors indicated different possibilities to explain this but in my view, they should be amplified, mainly because amino acid racemization is genus-dependent.

Minor suggestions:

Line 41: spelling of lithostratigraphic Line 121. The forams were oven-dried (4 hours at 30 °C). This heating may have produce an increase in racemization. Line 124. Why Core PS92/39-2 was only sampled for N. pachyderma? Line 127. Some samples needed 4h of immersion in H2O2 (instead of 2h) for removing the organic matter. Did the authors find any differences between sub-samples of the same level with these two different immersion times? Lines 152-156. I understand that these subsamples came from the same level. Line 170. Do you have any explanation for the high percentage (53%) of rejected samples in core 39-2? Lines 197-199. The authors observed that the extent of racemization in N. pachyderma samples was lower in PS92/39-2 below 3 m than in the other two cores. This core was drilled at a higher depth than the

other two, and in a northeastern position. Do they have information about temperature conditions or water currents which may have produced such changes? Indeed, the sedimentation rate of core 39-2 differed from the other two below 3-4 m (Fig. 3) Lines 211-215. The authors indicate that they observe stratigraphically reversed D/L Asp values in C. neoteretis samples from levels 3.12 and 4.45 of core PS92/45-2. However, it seems that one of these levels falls out the covariace trend of Asp and Glu acid D/L values. Line 232. I would not say that "dissimilar AAR rates between samples of comparable ages from different cores may originate from differences in sedimentation rates between the cores".

---

## Referee Comment (RC3) · Anonymous Referee #3 · 4 Sep 2019

This is an interesting work examining the nuances in interpreting amino acid racemization data for Quaternary foraminifera from the Arctic realm. The manuscript is generally well-written. Some minor comments are listed below: Line 39 - Plethora - word choice - seems extreme 47 - sample mass instead of size 66 - compared with 120 - although of admittedly of minimal effect on racemization, I was still surprised that the samples were oven dried - just a general principle 132 - upper case L in microlitre 133 why hydrolyse for free amino acids? - they should be available anyway even at low diagenetic temperatures 160 - in terms of the criterion, it is not necessarily so 217 coring process - is it possible for some younger foraminifers to more down the side of the core? Perhaps comment on exactly where in the core the samples were collected - central portion? 239 - word choice ' rough estimate' - approximation or first order assessment? I also

feel that some supporting discussion is merited on the validity of the rejection criterion L-ser>L-asp >0.8

---

## Author Comment (AC1) · 1 Oct 2019

We appreciate the time the reviewer took to review the manuscript and thank the reviewer for their constructive comments. We have tried to address them all and feel that they have improved the manuscript.

The reviewer's comments are in bold italics, followed by our responses. A revised manuscript with tracked changes is also included.

**Referee 1**

*"The study present amino acid data from three sediment cores raised from the eastern slope of the Yermak Plateau in the Arctic Ocean. Such studies have the possibility both to contribute in efforts to establish a reliable chronology and in efforts regarding identifying variability in bottom temperatures, both issues have been challenging in the Arctic ocean. The paper is well written and present an important set of data and deserve to be published. However, the discussion could be strengthen and the paper could be of larger general interest if the authors in a moderate to minor revision consider the following issues:*

1. *How good is the presented chronology beyond radiocarbon? Give some of the assumption it is based on and also the challenges in using a lithological/magnetic parameters in correlation in this geological setting. The age model for the older part of the core with the longest record seems reasonable, however the actual correlation seems not very convincing?"*

   The correlation between the records, and the original age models for these cores were recently presented by Wiers et al. (2019). They used the bulk density, magnetic susceptibility, and coarse fraction contents to provide an excellent correlation for the three cores. In the revised manuscript we have provided an additional figure (now Figure 3) that summarises this lithostratigraphic correlation on the original depth scales of the cores. We also altered the manuscript for clarification. We feel that this will remove some of the reservations of the reviewer, and for other readers, as to the robustness of the correlation.

   In addition to the solid inter-core correlation, the age models for these cores were also developed by Wiers et al. (2019). In the manuscript we describe how this was done, (i.e. primarily by correlation of environmental magnetic data to the global benthic $\delta^{18}O$ curve of Lisiecki and Raymo (2005), but constrained by some radiocarbon dates, and oxygen and carbon isotopes on foraminifera). Wiers et al. (2019) estimated the uncertainty for the correlation with the global benthic record of Lisiecki and Raymo (2005) for the cores to be approximately ± 6 kyr. The uncertainty takes into account bioturbation, and an estimate of the uncertainty in positioning tie points, and potential phase differences between the global benthic and the regional $k_{ARM}/k$ signals.

   In this paper, we have actually improved this error estimate by applying a Monte Carlo algorithm to more rigorously provide a measure of the uncertainty that is inherent to the correlation-based age models. This method is described in the manuscript and illustrates the higher level of uncertainty in the age models for the older part of the records.

2. *"IRD. In discussing ratios not in stratigraphic order, the authors should also discuss the possibility of reworking by sea ice and ice berg. Presentation of an IRD curve and some biostratigraphic data, if available, could be useful for this discussion."*

   The reviewer raises an interesting point. We referred to sediment mixing as a possible explanation for decimetre scale reversals in AAR rates. Naturally, several processes can be liable for sediment mixing, including bioturbation and deposition of reworked material entrained in sea ice and scouring of the seabed by icebergs. However, all of these coring sites were located on regions (or depths) of the Yermak Plateau that were free from iceberg scouring. We know this from the shipboard parasound and multibeam data. Furthermore, the

lithostratigraphic correlation between these records, precludes the possibility of large erosional events that could have removed or reworked sediments.

We have altered the text to include this explanation in the discussion.

3. *"Hydrography. The authors should present the main watermasses in the Arctic ocean basin and give the location of the cores relative to these. In the discussion of the diageneic temperatures it should be focused on possible changes in the deep water temperature and an evaluation of how this could influence on the amino acid diagenesis."*

We now include a new figure (now Figure 2) of potential temperature and salinity profiles obtained during the TRANSSIZ expedition from CTD stations nearest to the coring sites, and show where the cores are approximately located relative to the various water-masses. The following text to explain this has been added to the manuscript (2.1 Sample material and study area):
"The cores were recovered from water depths between 915 and 1464 m – in the Upper Polar Deep Water – far below the core of the warmer Atlantic layer, which typically sits between depths of 200 and 500 m (Jones, 2001). Potential temperature and salinity profiles from nearby CTD stations illustrate the relationship between the coring sites and the local water masses (Figure 2)." A new caption was introduced to describe the figure. Figure 1 was also updated.

In the revised manuscript we have not used our data-set to investigate possible changes in water mass temperatures in the past. We should emphasise that the calculation of effective diagenetic temperatures was performed as a rudimentary investigation in order to suggest a possible explanation for the observed differences in D/L values. For a complete assessment of the integrated post-depositional temperature history of the samples, the racemization kinetics and temperature sensitivity of the rate of AAR in *N. pachyderma* need to be formally established (which require laboratory heating experiments). Unfortunately, these were beyond the scope of this manuscript.

The over-reaching aim of the study was to test two fundamentally different empirical equations for calculating ages of *N. pachyderma* using AAR. We feel that the manuscript sufficiently answers this question. There is of course, further work that could now be done to look at more subtle influences on the AAR rates. This is touched upon in the discussion regarding the influence of temperature and/or sedimentation rates, but a more rigorous analysis would require higher resolution sampling of these cores. Now that we have established which age equation is applicable to the Yermak Plateau, these additional questions – highlighted by the reviewer - can be addressed as a follow-on project in the future.

4. *"In the discussion of the wider implication of the data it should be a more focused on published data on the same species from i basins with possibly different temperature histories,and investigate of the amino cid data could contribute to a discussion of changing deep water temperatures related to the climatic variability and locus of deep water formation ( early attempt on this in Sejrup and Haugen 1992)."*

Published studies on amino acid racemization in *N. pachyderma* from different basins, which have an established temperature history and utilise aspartic and glutamic acids are scarce. Comparison with earlier works is complicated, since most of the early works focused on the racemization of isoleucine, thus for a meaningful comparison with later studies, A/I values would need to be initially converted to equivalent D/L values (reverse phase). Whitacre et al. (2017) showed that this was straightforward for gastropod, bivalve, eggshell and whole rock samples, however, no studies assessed such conversion for foraminifera.

Again, the main aim of the study was to first address the larger question concerning whether the proposed 'Arctic' or 'Global' empirical age equations for AAR are applicable on the

Yermak Plateau. Now that we have addressed this, future work can focus on applications to ascertain temperature histories and paleoceanographic changes. Two main obstacles, both described in the text, prevent us from doing this: 1) We do not have measured geothermal gradients for the core sites, and 2) In the discussion we highlight the fact that sedimentation rate may exert as important a control on AAR rates as changing bottom water temperatures. This is something that needs to be tackled in a more detailed study in the future.

We have altered the text of the Discussion to clarify the above.

5. *"Background is generally good, however, it should be mentioned the large early work of D.L. Clark and others who in a number of papers suggested uniformly extremely low sedimentation rates in the Arctic ocean based mostly on interpretation of magnetic reversals. Amino acid data where actually the basis for some of the early challenging of this framework (Xuan and Channell 2010 intro )."*

We have altered the text to say: "Amino acid racemization (AAR) geochronology was first applied specifically to Arctic Ocean sediments in the pioneering studies of Sejrup et al. (1984), and later that of Macko and Aksu (1986). Sejrup et al. (1984) used AAR results to challenge the prevailing view of slow (mm/kyr) sedimentation rates (e. g. Clark, 1970) in the Arctic, and warned that the existing age interpretations derived from bio- and magnetostratigraphy could be substantially flawed. On the other hand, Macko and Aksu (1986) found that AAR chronology of sediments from the Alpha Ridge supported the accepted ages established by the use of these two dating techniques, thus, supporting arguments for slow sedimentation rates in this region."

**Additional changes**

We have renumbered the figures in the manuscript to reflect the introduction of the new figures. The reference list in the manuscript has been updated to include the newly cited literature (Clark, 1970 and Jones, 2001).

**References**

Whitacre, K. E., Kaufman, D. S., Kosnik, M. A., Hearty, P. J.: Converting A/I values (ion exchange) to D/L values (reverse phase) for amino acid geochronology, Quaternary Geochronology 37, 1-6, https://doi.org/10.1016/j.quageo.2016.10.004, 2017.

[revised manuscript text omitted]

---

## Author Comment (AC2) · 1 Oct 2019

We appreciate the time the reviewer took to review the manuscript and thank the reviewer for their constructive comments. We have tried to address them all and feel that they have improved the manuscript.

The reviewer's comments are in bold italics, followed by our responses. A revised manuscript with tracked changes is also included.

**Referee 2**

*"The authors present an interesting study about the amino acid racemization extent in the planktic foraminifera Neogloboquadrina pachyderma and the benthic species Cassidulina neoteretis collected at some depths in 3 cores drilled in the Yermak Plateau, in the Artic Ocean. The results obtained here contribute to increase the knowledge of amino acid racemization in foraminifera species, especially for the establishment of a reliable chronological model for the Artic, and the processes that may affect D/L values. In my view, the this study is of general interest, and the data set is important. Therefore, it is suitable to be published in Geochronology after minor revision:"*

1. *"In my view the authors should amplify the discussion regarding the influence of temperature and other factors in the D/L values that they observed. Planktic forams are subjected to marine currents and may remain in the water column for some time (hundreds?). This may produce the accumulation of tests with different ages in the same layer (time-averaging) and that these tests may have been racemized at different rates depending on original location, the place of the water column, etc. In fact, the authors indicated that "Lougheed et al. (2018) recently highlighted the large heterogeneity in the age distribution of foraminifera obtained from discrete depth intervals using 14C dating of single foraminifera". Moreover, the cores were drilled at different positions with marked diverse depths. Also, it has to be considered that the Yerkman Plateau is located in an area with interaction between Artic and Atlantic waters. The dominant colder or warmer currents may have affected the racemization rates. Do they authors have information about temperature gradients or water currents in the area?"*

Sinking velocities of *N. pachyderma* (80–144 μm) are in the range of 66 – 163 m day$^{-1}$ (Gyldenfeldt et al., 2000; Schiebel and Hemleben, 2000), implying that the tests spend no more than a few months in the water column prior to reaching the seafloor. This period is not sufficiently long to significantly influence racemization rates, considering that Arctic water masses are cold (< 3ºC).

Given the close geographical proximity (max. 130 km distance) and similar modern bottom water (–0.25, –0.43 and –0.76 ºC) temperatures reported for the three coring sites, we assumed that past bottom water temperature variability among the sites was likely minimal. We have not changed this assumption, but now include CTD profiles (Figure 2) from the nearest stations taken during the TRANSSIZ cruise. This does allow us to discuss the possibility of changing water masses, or water mass temperatures to some degree. We do not know, however, if and how much these temperatures changed over glacial cycles between the sites. We also emphasise that the calculated effective diagenetic temperatures can only serve as a basic, first order assessment to explain the offset between known-age-equivalent D/L values at the coring sites. Accurate palaeotemperature calculations are challenging due to high intra-sample variability and also require assessment of the racemization kinetics in *N. pachyderma*, both of which should be addressed in future studies. Furthermore, as we pointed out in our responses to the first reviewer, in addition to potentially varying bottom-water paleotemperature we also need to consider 1) geothermal gradients at the coring sites and 2) influence of sedimentation rates on AAR. We have altered the text of '4.3 Paleotemperature and other possible effects' to clarify this.

2. *"The results reveal an good match with the age equations of Kaufman et al. (2013) calculated using diverse species from sites of the Artic, Atlantic and Pacific Oceans (with different environmental contidions, e.g. temperatures). However, the age model of Kaufman et al. (2008), calculated for N. pachyderma of the Artic Ocean was not applicable here. The authors indicated different possibilities to explain this but in my view, they should be amplified, mainly because amino acid racemization is genusdependent."*

The differences predicted by the two calibrated age equations are indeed interesting and require further assessment. We obtained better agreement between AAR predicted ages on the Yermak Plateau with the mixed species global curve than with the *N. pachyderma* specific Arctic curve. This finding provides further evidence that the central Arctic data are anomalous, and the Arctic-specific equation needs to be re-visited. However, there are many possible explanations (e. g. differences between long-term rates of racemization in the Arctic Ocean and other oceans, diagenetic processes at the Yermak Plateau and other areas of the Arctic, issues with sample handling, or errors in the existing age models of central Arctic sediments) to account for such differences, we did not have the means to consider all of these options within this study. Now that we know that the global equation works in the cold water Arctic setting of the Yermak Plateau, we are in a position to investigate possible causes for the higher rates of racemization observed in *N. pachyderma* from the central Arctic Ocean.

*"Minor suggestions:*
*Line 41: spelling of lithostratigraphic Line 121. The forams were oven-dried (4 hours at 30 _C).*
*This heating may have produce an increase in racemization. Line 124. Why Core PS92/39-2 was only sampled for N. pachyderma? Line 127. Some samples needed 4h of immersion in H2O2 (instead of 2h) for removing the organic matter. Did the authors find any differences between sub-samples of the same level with these two different immersion times? Lines 152-156. I understand that these subsamples came from the same level. Line 170. Do you have any explanation for the high percentage (53%) of rejected samples in core 39-2? Lines 197-199. The authors observed that the extent of racemization in N. pachyderma samples was lower in PS92/39-2 below 3 m than in the other two cores. This core was drilled at a higher depth than the other two, and in a northeastern position. Do they have information about temperature conditions or water currents which may have produced such changes? Indeed, the sedimentation rate of core 39-2 differed from the other two below 3-4 m (Fig. 3) Lines 211-215. The authors indicate that they observe stratigraphically reversed D/L Asp values in C. neoteretis samples from levels 3.12 and 4.45 of core PS92/45-2. However, it seems that one of these levels falls out the covariace trend of Asp and Glu acid D/L values. Line 232. I would not say that "dissimilar AAR rates between samples of comparable ages from different cores may originate from differences in sedimentation rates between the cores".*

Line 41. We have corrected this spelling mistake.

Line 121. Oven drying the samples at 30ºC for a few hours does not substantially influence the extent of racemization in the samples. Laboratory heating experiments of *N. pachyderma* (see Kaufman et al. 2013, Table 2.) show that D/L values of aspartic acid increase by ~0.0006 over a 5–day period when the samples are subjected to 80ºC. All oven drying was performed at 30ºC for 4 hours, i.e. at much lower temperatures and shorter duration, resulting in an insignificant increase in D/L values.

Line 124. An initial pilot study targeted core 39-2, and aimed to look only at *N. pachyderma* and the influence of subsample size on intra-sample variability. No significant difference was found between subsamples containing 10 or 20 individuals. In a further effort to reduce intra-sample variability, the second round of sampling included the additional benthic species. We did not go back to core 39-2 and sample it for *C. neoteretis*.

Line 127. No differences were observed between subsamples of the same level with different immersion times.

Lines 152–156. We have now clarified this section. Indeed, for the comparison between subsamples of different sizes (i. e. 10 or 20 tests), the subsamples were taken from the same depths. This information was mentioned in the supplementary document, but we have now incorporated it in the main text of the manuscript.

Line 170. We do not have an explanation for this. It was in part why we thought that for the second round of sampling we should target another common species to provide a comparison. However, in the end, we had to reject far fewer of the *N. pachyderma* samples in cores 45-2 and 54-1. The reason for the high rejection rate (42 %) in the first batch of samples form core 39 remains a bit of a mystery.

Lines 197–199. We do not have any information on site specific geothermal gradients or paleo-water mass conditions that could help explain the apparently different AAR rates. We think that this is an important observation that needs to be investigated in future work. In addition to changes in the effective burial temperature (which can be influenced by bottom water temperatures and geothermal gradients), sedimentation rates may be an underexplored but important influence on AAR in foraminifera. Core 39-2 had the highest sedimentation rate. Lower rates of racemization coupled with high sedimentation was observed in previous AAR studies (e. g. Hearty et al., 2004). See also answer to Question 1.

Lines 211–215. Samples from both of these stratigraphic levels were assessed by the same rejection criteria. In both samples D/L Asp and Glu values covaried, but as we only had three sampled depths from this core, the trend of covariance of D/L Asp and Glu is poorly constrained, and we did not feel comfortable rejecting this data point.

Lines 232. We have altered the text to say: "Dissimilar D/L Asp and Glu values between samples of comparable ages from different cores…"

[revised manuscript text omitted]

Figure 3

[Figure]

Figure 4

[Figure]

Figure 5

[Figure]

Accepted data

Rejected data:
○ L-Ser / L-Asp > 0.8
◉ Non-covarying
◉ D/L Asp or Glu not within ± 2σ of the mean of the rest
○ Subsample damaged during analysis

Figure 6

*Neogloboquadrina pachyderma*

[Figure]

- ● PS92/39-2
- ■ PS92/45-2
- ◆ PS92/54-1

*Cassidulina neoteretis*

[Figure]

Figure 7

[Figure]

Figure 8

*Neogloboquadrina pachyderma*

[Figure]

*Cassidulina neoteretis*

Figure 9

[Figure]

*Neogloboquadrina pachyderma*

R²=0.65
p = .000032

*Cassidulina neoteretis*

R²=0.82
p = .000046

[Figure]

● PS92/39-2
■ PS92/45-2
◆ PS92/54-1

**Tables**

**Table 1. Sediment cores analysed in this study.**

[revised manuscript text omitted]

---

## Author Comment (AC3) · 1 Oct 2019

We appreciate the time the reviewer took to review the manuscript and thank the reviewer for their constructive comments. We have tried to address them all and feel that they have improved the manuscript.

The reviewer's comments are in bold italics, followed by our responses. A revised manuscript with tracked changes is also included.

**Referee 3**

*"This is an interesting work examining the nuances in interpreting amino acid racemization data for Quaternary foraminifera from the Arctic realm. The manuscript is generally well-written. Some minor comments are listed below: Line 39 - Plethora - word choice - seems extreme 47 - sample mass instead of size 66 - compared with 120 - although of admittedly of minimal effect on racemization, I was still surprised that the samples were oven dried - just a general principle 132 - upper case L in microlitre 133 why hydrolyse for free amino acids? - they should be available anyway even at low diagenetic temperatures 160 - in terms of the criterion, it is not necessarily so 217 coring process – is it possible for some younger foraminifers to more down the side of the core? Perhaps comment on exactly where in the core the samples were collected - central portion? 239 - word choice' rough estimate' - approximation or first order assessment? I alsofeel that some supporting discussion is merited on the validity of the rejection criterion L-ser>L-asp >0.8."*

Line 39. We have altered the text to say: "Dating Quaternary marine sediments from the Arctic Ocean has been a long-standing problem, and a number of studies (e.g. Backman et al., 2004; Stein, 2011; Alexanderson et al., 2014) highlight the challenges of establishing firm chronologies for these sediments."

Line 47. 'Sample size' was used to characterise the number of foraminifera tests (hundreds) required for each analysis performed on gas or ion-exchange chromatography in the early studies. This limitation was overcome by the introduction of the reverse-phase chromatography (Kaufman and Manley, 1998), which allowed the analysis of even one single test, thus reducing the sample size necessary for analyses.

Line 66. We have changed the text to say: "rates of racemization are anomalously high compared with other ocean basins".

Line 120. As we argued to the second reviewer, oven drying the samples at 30ºC for a few hours does not substantially influence the extent of racemization in the samples. Laboratory heating experiments of *N. pachyderma* (see Kaufman et al. 2013, Table 2.) show that D/L values of aspartic acid increase by ~0.0006 over a 5–day period when the samples are subjected to 80ºC. All oven drying was performed at 30ºC for 4 hours, i.e. at much lower temperatures and shorter duration, resulting in an insignificant increase in D/L values.

Line 133. Most commonly, amino acid geochronology relies on the entire population of amino acids, i.e. both intra- and inter-crystalline, in the sample (e. g Mitterer, 1993; Penkman et al., 2008). Hydrolysis is necessary to release all peptide-bound amino acids to recover a larger pool of amino acids, which is necessary when analysing such small sample sizes. Analysing just the naturally free amino acids would require much larger samples because the concentration is much lower.

Line 160. For consistency we adhered to previously published studies, which used the L-Ser/L-Asp > 0.8 rejection criteria as a way to screen for possible contamination by modern amino acids (Kaufman, 2006; Kosnik and Kaufman, 2008; Kaufman et al., 2008; Kaufman et al., 2013). Indeed, many subsamples that exhibited L-Ser/L-Asp values >0.8 also deviated from the trend of the positive covariance between D/L Asp and D/L Glu, or would have otherwise

been rejected for falling outside ±2σ of the sample mean. These rejection criteria were implemented to emphasise the central tendency of the data, and to remove outliers.

Line 217. We have included the following sentence in section '2.3 Sample preparation and analytical procedure' for clarification: 'The sediment slices were taken from u-channel samples that had been used for paleomagnetic measurements, implying that the samples came from the central parts of the cores.'

Line 239. We have altered the text to say: "Here we provide a rudimentary estimate of the difference in the effective diagenetic temperatures required to account for the offset between known-age-equivalent D/L values between the coring sites."

[revised manuscript text omitted]

— PS92/39-8    *PML* - Polar Mixed Layer
— PS92/46-2    *AL* - Atlantic Layer
— PS92/55-1    *UPDW* - Upper Polar Deep Water.

Figure 3

[Figure]

Figure 4

[Figure]

Figure 5

[Figure]

●     Accepted data

Rejected data:
○     L-Ser / L-Asp > 0.8
○     Non-covarying
○     D/L Asp or Glu not within ± 2σ of the mean of the rest
○     Subsample damaged during analysis

Figure 6

*Neogloboquadrina pachyderma*

[Figure]

- PS92/39-2
- PS92/45-2
- PS92/54-1

*Cassidulina neoteretis*

[Figure]

Figure 7

[Figure]

Figure 8

*Neogloboquadrina pachyderma*

[Figure]

Figure 9

[Figure]

[Figure]

585

**Table 1. Sediment cores analysed in this study.**

[revised manuscript text omitted]